# Radiomics for the Prediction of Pathological Complete Response to Neoadjuvant Chemoradiation in Locally Advanced Rectal Cancer: A Prospective Observational Trial

**DOI:** 10.3390/bioengineering10060634

**Published:** 2023-05-24

**Authors:** Liming Shi, Yang Zhang, Jiamiao Hu, Weiwen Zhou, Xi Hu, Taoran Cui, Ning J. Yue, Xiaonan Sun, Ke Nie

**Affiliations:** 1Department of Radiation Oncology, Sir Run Run Shaw Hospital, Zhejiang University School of Medicine, Hangzhou 310019, China; 2Department of Radiation Oncology, Rutgers–Cancer Institute of New Jersey, Rutgers-Robert Wood Johnson Medical School, 195 Little Albany St., New Brunswick, NJ 08903, USA; 3Department of Radiology, Sir Run Run Shaw Hospital, Zhejiang University School of Medicine, Hangzhou 310019, China

**Keywords:** MRI radiomics, pathological complete response, locally advanced rectal cancer, prospective trial

## Abstract

(1) Background: An increasing amount of research has supported the role of radiomics for predicting pathological complete response (pCR) to neoadjuvant chemoradiation treatment (nCRT) in order to provide better management of locally advanced rectal cancer (LARC) patients. However, the lack of validation from prospective trials has hindered the clinical adoption of such studies. The purpose of this study is to validate a radiomics model for pCR assessment in a prospective trial to provide informative insight into radiomics validation. (2) Methods: This study involved a retrospective cohort of 147 consecutive patients for the development/validation of a radiomics model, and a prospective cohort of 77 patients from two institutions to test its generalization. The model was constructed using T2-weighted, diffusion-weighted, and dynamic contrast-enhanced MRI to understand the associations with pCR. The consistency of physicians’ evaluations and agreement on pathological complete response prediction were also evaluated, with and without the aid of the radiomics model. (3) Results: The radiomics model outperformed both physicians’ visual assessments in the prospective test cohort, with an area under the curve (AUC) of 0.84 (95% confidence interval of 0.70–0.94). With the aid of the radiomics model, a junior physician could achieve comparable performance as a senior oncologist. (4) Conclusion: We have built and validated a radiomics model with pretreatment MRI for pCR prediction of LARC patients undergoing nCRT.

## 1. Introduction

The standard-of-care treatment for locally advanced rectal cancer (LARC) is neoadjuvant chemoradiation therapy (nCRT) followed by total mesorectal excision (TME) [1]. Despite the fairly standard treatment protocol, responses vary, with only 15–25% of patients achieving a pathologic complete response (pCR) [2]. For those who achieve pathologic complete response, the use of TME has been questioned due to inferior quality of life and no improvement in overall survival. Instead of TME, the organ preservation strategy known as “wait-and-watch” has been proposed [3]. The major challenge is to identify appropriate approaches for effective early prediction. Magnetic resonance imaging (MRI) is considered to be the most accurate modality for rectal cancer staging, and it is routinely used in clinical practice [4]. However, its role in the prediction of the treatment response is unclear [5,6]. The main challenges for widespread clinical use of MRI are the intrinsic limitation of imaging itself for identifying treatment-reduced changes and observer dependency.

Recently, an increasing amount of research has been conducted to explore the role of radiomics in nCRT response prediction, with the aim of providing additional biomarkers for better management of LARC [7,8,9,10]. Shin et al. built a radiomics model from post-treatment MRI features and yielded an area under the curve (AUC) of 0.82 [11]. However, the post-treatment analysis did not provide direct guidance of early intervention. Consequently, others have tried using pretreatment image features to identify the pathologic complete response earlier. Yi et al. designed a radiomics signature from T2w MRI to predict pCR and reported prediction power with an AUC of 0.91 [12]. DiNapoli et al. extended the work with multicenter datasets but only achieved an AUC of 0.73 for internal and 0.75 for external validation [7]. Zhou et al. applied 16 radiomic features from multi-parametric MRI to predict pCR and achieved an AUC of 0.82 [13]. However, when comparing these studies, all were conducted retrospectively, and none of the proposed radiomics models shared a single feature. In addition, only the diagnostic index (AUC) was reported. An intuitive understanding of the “black-box” nature of the radiomics model and how to guide clinical decisions has not been provided. The transparency and reproducibility of these studies have led to translational gaps between research and clinical practice. Recently, a new guideline was released, i.e., the Transparent Reporting of a Multivariable Prediction Model for Individual Prognosis or Diagnosis (TRIPOD) statement [14], from the SPIRIT-AI and CONSORT-AI consortia. A multicenter prospective validation is highly recommended to prove model generalization.

In light of this, the aim of this study is to design and validate a prognostic radiomics tool using multi-parametric MRI for pCR prediction and to compare the predictive power with treating physicians’ readings on a multi-institutional prospective clinical trial (ChiCTR2000029101). A cross-comparison of the treating physicians’ performances with and without the aid of the radiomics model was conducted to assess the model’s potential role in assisting clinical decisions. Moreover, the correlation between the predictive features and their corresponding radiological meanings were re-evaluated by radiologists, which could facilitate a smoother transition of the research tool from a “black box” to a clinical process.

## 2. Materials and Methods

### 2.1. Study Population

This institutional review board (IRB)-approved study involved a retrospective observational cohort for the development and validation of the radiomics model, and a multicenter prospective testing cohort to assess the generalization of the model. The inclusion and exclusion criteria for each cohort are shown in Figure 1 and detailed explanations are provided in Appendix A. For the observational cohort, 320 consecutive LARC patients (cT3–cT4 tumor stage without distant metastasis) treated at Institution I (Sir Run Run Shaw Hospital) between January 2010 and July 2015 were reviewed. In total, 147 patients were included in the analysis. Another 313 patients, enrolled in a prospective trial (hiCTR2000029101) from two centers, i.e., Institution I (Sir Run Run Shaw Hospital) and Institution II (Hangzhou Xia Sha Hospital) were collected. Among these, 77 eligible patients were included in the testing cohort. The details of the patients’ clinical characteristics are presented in Table 1.

To minimize image acquisition variation, only patients scanned with 3.0T GE scanners (Signa, Innsbruck, Austria) from both centers were used for analysis. The multi-parametric MRI was performed two weeks before the nCRT. The image sequences included T2-weighted (T2w) images, dynamic contrast-enhanced (DCE)-MRI, and diffusion-weighted images (DWIs) with b-values of 0 and 800 s/mm^2^. The detail acquisition parameters are listed in Appendix B. All patients received 5-FU based concurrent chemoradiation (nCRT) with 45–50Gy, and TME was performed 6–8 weeks after. The evaluation of surgical specimen was reviewed by two experienced gastroenterology pathologists and a third expert pathologist was responsible for the final decision in the case of a disagreement. Then, the patients were separated into two groups as pathological complete response (pCR as TRG 0) or non-pCR (as TRG 1–3). Details regarding the treatment protocols and pathology readings are provided in Appendix C.

### 2.2. Radiomics Model Evaluation

A radiation oncologist (L.S. with 8 years of gastrointestinal MRI experience) was responsible for the delineation of the tumor region of interest (ROI) on each individual sequence, under the mentorship of an expert radiologist (X.H. with 15 years of experience) by using the MIM Maestro^®^ software (MIM software Inc., Beachwood, OH, USA). When uncertainty existed concerning the treated tumor region, the area was not included in the segmentation.The delinationprocess is inllustated in Appendix D.

The entire radiomic process is shown in Figure 2. Radiomic features consisting of shape and first-order/second-order texture-based measures were computed with PyRadiomics version 2.1.2 (https://pyradiomics.readthedocs.io, (accessed on 1 May 2022)). The image preprocess and feature calculation are illustrated in Appendix E. Needless to say, to assess the dataset homogeneity across institutions, batch effects were evaluated using the ComBat method. Batch effects can manifest as non-biological variations in data due to inconsistencies in sample processing, handling, or instrument calibration, which, if unaddressed, can lead to incorrect conclusions. While conducting an assessment using the ComBat method, we observed that the chosen radiomic features were consistent with those identified without any ComBat adjustment. This suggested that potential batch effects did not substantially impact the feature selection process in this study. Overall, a total of 428 parameters were calculated for each case. A support vector machine (SVM) was trained to separate the patients by their treatment outcomes after balancing the number of patients in different response categories by using the synthetic minority oversampling technique [15]. Then, a radiomics score with the probability of being pCR was computed again using the SVM classifier. For comparison, a clinical model which incorporated all clinical data such as age, gender, TNM staging, tumor location, and treatment option, but not MRI data, was built using a random forest classifier. A 5-fold cross-validation was performed within the observational set and repeated 5 times with average results reported. The predictive performance of the primary model was further assessed on the prospective testing cohort.

### 2.3. Qualitative Evaluation

The MR images from the prospectively testing cohort were evaluated independently by two radiation oncologists (L.S. and W.Z. with 2 years of experience respectively). Before the reading sessions, both readers were trained by a radiologist’s group using a separate patient cohort with data from 60 patients (not included in this study analysis). They were also provided published recommendations and guidelines for MRI interpretation of LARC [5,6,7,16,17,18,19]. During the reading session, they were given all image sequences, the primary TNM staging, and other clinical information, but they were blinded to patient surgical pathological results. Specifically, for the senior oncologist, his readings were performed under the consultation of an experienced radiologist (X.H. with 15 years of experience).

Based on the image readings and the corresponding baseline clinical characteristics, the assessments were categorized based on probability as: I, highly likely a pCR; II, probably a pCR, III, probably a non-pCR; and IV, highly likely a non-pCR. Cases in categories I–II were labeled as pCR and the remaining cases were labeled as non-pCR. A total of 6 reading sessions were conducted for each reader. Patients’ cases were anonymized, re-labeled, and shuffled before being presented for assessment. Reading sessions #1–#3 were performed without the aid of the radiomics model, and two months later, reading sessions #4–#6 were conducted with the aid of the radiomics model output. In each reading session, all cases were reviewed within a week with each session separated at least a week apart.

### 2.4. Statistical Analysis

The comparison studies were constructed to evaluate: (1) the predictive power of the radiomics model vs. the clinical model vs. the readers’ study; (2) the consistency of treating physicians’ readings without vs. with the aid of the radiomics model; and (3) the predictive power of both readers without vs. with the aid of the radiomics model. The sample size of the prospective cohort was calculated based on the accuracy of the radiomics model for predicting pCR in the retrospective cohort with an assumption of 15–20% pCR incidence rate and a goal of α = 0.05, 1 − β = 0.95 using SPSS 26.

The intra- and inter-reader consistency was analyzed using the weighted kappa criteria (κ = 0.00–0.20, poor; 0.21–0.40, fair; 0.41–0.60, moderate; 0.61–0.80, good; 0.81–1.00, excellent agreement). The performance for identifying pCR was derived and compared using a range of classification metrics such as the area under the curve (AUC), accuracy, sensitivity, specificity, positive predictive value (PPV), and negative predictive value (NPV). The optimal cutoff values of the radiomics score were determined by maximizing the Youden index in the training set. Two distinct statistical procedures were conducted to compare the models’ performances: the DeLong test for AUC comparisons and the exact Cochran–Mantel–Haenszel (CMH) test for examining the other prediction indices (accuracy, sensitivity, specificity, PPV, and NPV). To perform the DeLong test, the ROC curves were initially generated for both models by plotting sensitivity against the false positive rate (1-specificity) across a range of decision thresholds. Subsequently, the DeLong test was utilized to determine if there were notable differences in the AUC values between the models. A statistically significant result would suggest a disparity in their capacity to differentiate pCR from non-pCR cases. To perform the exact CMH test, contingency tables were firstly constructed for each index followed by the CMH test statistics. The test is designed to analyze the association between categorical variables while accounting for potential confounding factors. In all cases, *p*-value ≤ 0.05 was considered to be statistically significant.

## 3. Results

### 3.1. Radiomics Model Evaluation

Figure 3 shows the prediction results for the training, validation, and testing cohort. The power analysis revealed that a sample size of 65–72 patients was needed, if assuming a pCR rate of 15–20%.

The radiomics model outperformed the clinical model in all three cohorts. Specifically, in the prospective testing cohort with 77 patients, the radiomics classifier demonstrated an AUC of 0.84 (95% CI 0.70–0.94) compared to an AUC of 0.69 (95% CI 0.62–0.77) for the clinical model. The predictive power was not significantly improved by the combination of clinical data with an AUC of 0.85 (95% CI 0.70–0.95). A total of 14 radiomic features were selected with their relative importance displayed in Figure 3d. The detailed radiomic feature results between the pCR group and the non-pCR group are listed in Appendix F; eight features were from DCE-MRI, three features were from T2w images, and three features were from DWIs, demonstrating that all three sequences were valuable for prediction. Most of the selected features were texture features indicating that internal heterogeneity was an important indicator for response. Among the top five selected features, three features were from DCE-MRI and two features were from T2w images. The ADC map generated from the DWIs showed its predictive power, but the significance level was lower compared to the other two sequences.

The intra- and inter-reader consistency for prediction of the pCR without and with the aid of the radiomics model are shown in Figure 4. The readers, regardless of their levels of experience, could only produce fair to moderate consistency even among their own assessments. In contrast, with the aid of the radiomics model, the inter-reader assessment consistency was significantly improved with good to excellent agreement, regardless of the experience levels.

This indicates, with the assistance of the radiomics tool, that an inexperienced reader may achieve performance comparable to that of a senior physician. Since readers’ bare readings did show good consistency, only the readouts from the last reading session in each round (Sessions #3 and #6) were used to compare with the radiomics model and the clinical model. The differences between qualitative and quantitative assessments were significantly different for the vast majority of the results, with *p*-values demonstrated in Figure 5. Especially, the specificity and PPV were significantly improved with the assistance of the radiomics model compared to the readers’ bare readings.

### 3.2. Understanding the Selected Radiomic Features

Furthermore, the model-determined radiomic features were compared between the pCR and non-pCR groups. The detailed interpretation of the selected features was attempted by looking into the cases having high and low radiomics scores. Here, we present four feature examples, two from DCE-MRI and two from T2w images.

The most important feature determined in the model was the texture measure from the gray level run length matrix (GLRLM) collected from the DCE-MRI. The feature long run low gray level emphasis (LRLGLE) measures the joint distribution of long run lengths with lower gray-level values, with a higher value (pCR case) indicating greater concentration of low gray-level values in the images. Figure 6a shows two case examples. The pCR case with a feature value of 0.91 vs. 0.21 for non-pCR, shows a larger portion of lower enhancement than the non-pCR case on the DCE-MRI (subtraction signal between the second phase and the first phase). Another representative feature is the neighborhood gray-tone difference matrix (NGTDM) coarseness, the third highest weighting parameter among all radiomics features. It is associated with image granularity with a higher value (pCR case) indicating a lower spatial change and a locally more uniform texture. Figure 6b shows two cases in which the pCR case presented a higher value (less contrast) compared to the non-pCR case. The overall observation was that the tumors representing lower enhancement signals at the initial enhancement phase and lower contrast were likely to respond better to the treatment regimen.

The radiomic features captured from the T2w images also played significant roles in assessing and predicting pCR. One example is the first-order image feature as histogram standard deviation (T2 SD) which directly measures the intensity variation on T2w images. Figure 7a shows two cases with the top row as the pCR which has lower signal as well as lower variation on the corresponding T2 images compared to the non-pCR case. The other example is the second-order feature as gray level co-occurrence matrix (GLCM) joint average. A homogeneous GLCM matrix means uniform signal distribution and returns a zero joint average. As illustrated in Figure 7b, the pCR case tends to have more homogeneous T2w signal (lower joint average) compared to the non-pCR case.

## 4. Discussion

Reliable and noninvasive early identification of the nCRT response is essential for guiding the optimal management of patients with LARC. To the best of our knowledge, this is the first study to compare an MR-based radiomics model with physicians’ descriptive evaluations on a prospective trial in an attempt to define early nCRT response in patients with LARC. We found that the physician’ assessments with multi-parametric MRI showed only fair to moderate inter- and intra-reader agreement. The predictive performance also showed limited agreement with pathological results. Instead, the radiomics model demonstrated higher predictive power, offering significantly better specificity and PPV. If assisted with the radiomics model, the physicians’ readings, regardless of the experience level in MR, could be improved with high consistency and much better matching with the pathology. Furthermore, based on the case review, the indication from the selected radiomic features demonstrated an inherent link between homogeneous appearance and less contrast with better nCRT response.

The observation of pCR has led to a paradigm shift in rectal cancer management. It has been reported that patients with pCR after nCRT, in general, have favorable oncological prognoses, with a 5-year disease-free survival rate reaching 83–95% [16]. Additionally, to avoid surgery-associated morbidity and impairment in quality of life, patients with complete response following nCRT are increasingly being offered watch-and-wait regimens or organ-sparing strategies, such as local excision [17]. To further increase the number of eligible patients for such organ preservation strategies, physicians are searching for (new) neoadjuvant treatments with higher organ-sparing potential than the current standard of care, as well as searching for potential biomarkers to predict pathological complete response before the surgery [18]. The benefit of avoiding extensive surgery is a reduction in the significant risk of perioperative morbidities, including bowel, sexual, and urinary dysfunction. Unfortunately, there is no established model or method for early prediction of a pathological complete response, while the pCR assessment can only be confirmed after surgery. Limited clinical parameters have been reported to be associated with pathological complete response, which include TNM staging, tumor size, tumor location, etc. Moreover, most of the parameters reported in the literature were conducted in research settings and have not been verified by clinical trials.

Due to a lack of established biomarkers that can help to determine neoadjuvant treatment effects, currently, a thorough analysis of medical images is the only dependable modality for determining response. This includes a longitudinal analysis of each patient as well as a cross-sectional evaluation at a specific point in time, leading to an exponential increase in physicians’ workloads. Furthermore, anatomical, and functional MRI has intrinsic limitations in determining radiation-induced effect [10]. For example, the extramural tumor invasion, a typical appearance in advanced stage, is difficult to identify due to spiculated low-intensity signal on T2w images [19,20]. Radiation-induced edema and inflammation also result in similar signals that resemble those of tumors on T2w images [21]. Human reading of these images is time-consuming and prone to various types of errors, even for experienced radiologists. Khwaja et al. proposed a novel multi-parametric MRI scoring system and two radiologists of different experience levels interpreted 64 patients’ images [22]. The diagnostic performance for identifying good responders was much improved, but the overall accuracy was still low compared to histopathological results (non to low agreement for the junior and senior radiologist, respectively). In another reader study with 24 patients, Hotker et al. reported that inter-reader agreement differed significantly and agreement using post-treatment images was even worse [23]. Since the post-treatment images did not offer valuable guidance for treatment strategy adaption and they were more error prone for reliable reading, in this work, we focused on a radiomics tool with pretreatment images. Additionally, the current cocktail regimen of cancer management requires multidisciplinary team efforts. It is expected that physicians who actively manage treatment, such as radiation oncologists who have less experience as radiologists interpreting images, would encounter more difficulties in evaluating an individual patient’s profile and making treatment adjustment suggestions. Therefore, a quantitative tool as presented here is urgently needed.

In recent years, there has been a rapid increase in research studies to support the role of radiomics in LARC for pCR prediction. Yi et al. built a radiomics signature from pretreatment T2w MRI and reported prediction power with an AUC of 0.91 for the prediction of pCR [12]. Dinapoli et al. extended the work with external validation but only achieved an AUC of 0.73 [7]. Liu et al. claimed a nearly perfect AUC of 0.98 using post-treatment T2w and ADC map [8]. Recently, Shin et al. performed a large-scale validation and confirmed the radiomics model from post-treatment T2w and ADC map, but only yielded an AUC of 0.82 [11]. Despite the encouraging yet variable results, the post-treatment analysis did not provide direct, useful guidance for treatment adjustment. It was, thus, a diagnostic process to link clinical complete response (CCR) to pCR instead of early prognosis prediction. The contouring of residual disease could be difficult on after-radiation images, to result in reliable radiomics model construction. A few other research studies have used pretreatment multi-parametric MRI to predict pCR, successfully yielding AUC values of 0.7–0.9 [9,10,24]. However, the clinical applicability was not clear since most of those studies had small sample sizes of not only retrospective nature but also primarily single-institute datasets. To overcome the limitations of the previous studies, following the recently released guidelines such as the TRIPOD statement by the SPIRIT-AI and CONSORT-AI consortia, we conducted a multicenter prospective validation study. In our study, it was found that the AUC for pCR prediction using pretreatment multi-parametric MRI could reach a high value of 0.84 (95% CI 0.70–0.94), suggesting the potential of generalization of this radiomics model in real clinic scenarios.

Another issue that hinders the clinical adoption of radiomics study is the “black-box” nature of the model. The recent development of the deep-learning algorithm has pushed the field towards increasing predictive power and further away from understanding the underlying relationships [25]. In this study, support vector machine (SVM) was used for classification and regression. Although it is a widely used machine learning algorithm due to its robustness to outliers and its ability to handle high-dimensional data, the interpretability is limited; it is difficult to interpret and explain compared to simpler models such as linear regression or decision trees. This lack of interpretability can be a concern when it is important to understand the relationships between features and predictions. To transform a radiomics analysis into an actionable process, instead of simply reporting the AUC value, we presented the radiomics outcome as the probability of pCR. Furthermore, we provided the values of key features of each individual case for physicians to review. As such, the relationships between the data-driven radiomics and readers’ visual characteristics could be linked to support informational insight for radiomics validation. It was found that tumors with lower initial enhancement and less contrast on DCE-MRI and a homogeneous approach on T2w MRI tended to respond better. These observations can be explained by the fact that aggressive tumors may have hypoxic oxygen supplies which result in heterogeneous signals on DCE-MRI. An additional derived correlation is that tumors with mucinous area that have higher signal intensity on T2w MRI typically had worse prognoses. Overall, these feature findings were correlated to the radiologists’ impressions regarding good responders vs. nonresponders, but further validations with larger prospective cohorts are needed to confirm their clinical use.

There are a few limitations in the current study. Firstly, the sample size was still limited, which has been a general concern for most radiomics-based studies. The model was developed using a retrospective cohort design, and although it was validated in a multicenter prospective cohort, it might still introduce bias and limit its generalization. We performed a power analysis. The sample size of the prospective cohort was calculated based on the accuracy of the radiomics model for predicting pCR in the retrospective cohort with an assumption of 15–20% pCR incidence rate and a goal of alpha = 0.05 and 1-beta = 0.95. The power analysis based on the retrospective observational cohort result revealed that a minimum of 65–72 patients were needed to achieve reasonable power. In the prospective cohort of the study, we collected 77 patients with the same inclusion and exclusion protocol to meet the power requirement. We also understand that the limited sample size may have impacted the statistical power and contributed to overfitting. In this study, the synthetic minority oversampling technique (SMOTE) was applied to overcome imbalanced class distribution. It is a conventional method for oversampling the minority class in a dataset with imbalanced class distribution. It has been adopted for processing of medical data [26]. It works by generating synthetic samples of the minority class by interpolating between existing samples. To avoid over-representation, we compared the performance of SMOTE and other methods, including random oversampling or cost-sensitive learning, etc. However, those methods introduced more bias with worse performances. To find the best hyperparameter and strategy to control overfitting in this study, we kept assessing the performance of the model on validation data to determine its generalizability through cross-validation. Yet, we acknowledge this limitation, and the future direction is to expand the study with a large multi-institutional prospective clinical trial design. By recruiting a diverse patient population from multiple medical centers, we can enhance the heterogeneity of our sample, reduce potential biases, and account for variations such as patient demographics. The prospective nature of the study will also allow us to follow patients in real time, enabling more accurate evaluation and refinement of the model based on temporal disease progression and patient outcomes.

Secondly, we acknowledge that the using a specific MRI protocol (3.0T GE scanners with T2w, DCE-MRI, and DWI sequences) in our study might have led to a unique set of radiomic features, which could be protocol specific. Variations in factors such as magnetic field strength, pulse sequences, and acquisition parameters across different MRI protocols may result in differences in image contrast, signal-to-noise ratio, and spatial resolution. Consequently, the model’s performance might vary when applied to images acquired using different protocols or from different imaging modalities. As such, we limited our study to those images captured using the same strength MRI scanners and identical image protocols to minimize the variations. However, to extend this model with more generalizations, a large population study is needed. By including a diverse set of imaging data, we can capture variations in radiomic features that are representative of real-world clinical settings. This will enable us to evaluate and refine the model’s performance across a wider range of imaging scenarios, thus, enhancing its applicability and usefulness for clinicians.

Thirdly, we recognize that factors such as gradient nonlinearity, magnetic field inhomogeneity, motion artifacts, and eddy current-induced distortions can result in discrepancies and artifacts within DWI images. It is essential to consider these factors when developing and validating our radiomics model to ensure reliable and accurate results. In the context of heterogeneity of magnetic field gradients, two main trends have been proposed in the literature to address these challenges. The first trend, introduced by Bammer et al., involves the coil tensor, which assumes that inhomogeneities arise only from gradient coils. This method takes into account the spatial distribution of the magnetic field gradients generated by the gradient coils and aims to correct the effects of these inhomogeneities on the DWI images [27]. By modeling and compensating for the contributions of gradient coil-induced magnetic field distortions, this approach helps to minimize systematic errors and to improve the accuracy of diffusion-weighted image analysis. The second trend, which is B-matrix spatial distribution (BSD) in DWI/DTI, employs diffusion tensor standards to determine the spatial distribution of the B-matrix, which is a function of the spatial distribution of field gradients [28]. The BSD-DTI method allows for the assessment of the actual distribution of field gradients, accounting for those originating from gradient coils as well as background, eddy currents, and radiation damping. By considering these sources of inhomogeneity, researchers can identify and correct systematic errors in DWI/DTI, resulting in more accurate image analysis and interpretation.

An additional limitation of this study is that the tumor regions of interest (ROIs) were delineated manually, through a consensus-based approach. A single experienced radiation oncologist manually outlined all the tumors under the guidance of another experienced radiologist. Despite a certain level of consistency, this method presents certain drawbacks as it is not possible to conduct an inter-reader agreement assessment. As a result, the reliability of the radiomics measures derived from these manually drawn ROIs may be compromised. To address these concerns and to enhance the reliability of the radiomics measures, it is essential to consider incorporating semi-automatic or fully automatic segmentation techniques for tumor ROI delineation. These advanced segmentation methods have the potential to reduce subjective errors and biases introduced by manual delineation, thus, improving the consistency and reproducibility of the tumor ROIs. Moreover, adopting semi-automatic or fully automatic segmentation approaches can significantly facilitate the use of radiomics measures in clinical practice. By streamlining the ROI delineation process, these methods can save valuable time and resources while ensuring that radiomics analyses are more reliable and robust. Ultimately, this would promote the integration of radiomics measures into clinical decision-making, potentially improving patient outcomes and advancing personalized medicine

In addition to expanding the study with a large multi-institutional prospective trial design, future direction is to expand the study by incorporating other prognostic factors, such as cytotoxic T-lymphocyte-associated protein 4 (CTLA-4). CTLA-4 inhibits T cell activation, which helps maintain immune homeostasis and prevent autoimmunity. Cancer cells can exploit this mechanism by upregulating CTLA-4 expression, thereby, suppressing the immune response against them. As a result, tumors can grow and metastasize without being recognized and eliminated by the immune system. This has led researchers to investigate the role of CTLA-4 in various cancers, including colorectal cancer. Several studies have demonstrated that CTLA-4 is indeed expressed in SW480 cells, albeit at varying levels depending on the experimental conditions and detection methods used. A study by Kamal et al. found that the gene expression of CTLA-4 was significantly upregulated in colorectal cancer (CRC) patients compared to a control group (*p* < 0.001). Individually, CTLA-4 showed 85% sensitivity in discriminating CRC patients from the control group (*p* < 0.001) [29]. Another study by Ghorbaninezhad et al. found that cytotoxic T-lymphocyte antigen-4 (CTLA-4) silencing in dendritic cells loaded with colorectal cancer cell lysate improved autologous T cell responses in vitro [30]. Nevertheless, investigating the expression and function of CTLA-4 in SW480 cells can provide valuable insights for evaluating the potential immunotherapies targeting CTLA-4, such as immune checkpoint inhibitors to colorectal cancer. Given the growing interest in radiogenomics, it is possible that future studies will investigate the relationship between radiomic features and specific immune-related genes, such as CTLA-4, in rectal cancer. This may help to identify noninvasive imaging biomarkers associated with gene expression, which could potentially be used to select patients who may benefit from radiation or targeted therapy.

To summarize, our study showed that physicians’ assessments for predicting pathological outcome based on pretreatment multi-parametric MRI had limited consistency and accuracy. However, their performances could be significantly improved with the aid of a radiomics model. This will be especially beneficial for radiation and medical oncologists who manage treatment but have limited experience in annotating MR images, as they can achieve comparable performances to senior radiologists for assessing each individual patient’s tumor profile at an earlier stage. Moreover, we further built a correlation of key features from the radiomics model and physicians’ interpretations, finding that a homogeneous appearance shown on DCE-MRI and T2-weighted images was linked to better treatment outcomes. This would also assist physicians by providing them with not only a black box-based score, but an intuitive understanding of what types of lesions may show better treatment outcomes. Overall, the developed radiomics model may facilitate the prediction of treatment responses in the early phase. For patients deemed to have a high probability of achieving a good response, an intensified regimen or boost radiation may be applied to maximize their chances of getting a complete response. On the one hand, this would be extremely beneficial for patients with cancer close to the rectum, for whom conservative surgery with a “wait-and-watch” strategy can be considered for sphincter sparing and preservation of urinary and sexual functions. On the other hand, for patients who show a low probability of receiving a good response in the early phase of treatment, modified nCRT with less toxicity and side effects can be considered.

## 5. Conclusions

In summary, the radiomics model based on pretreatment multi-parametric MRI showed good predicting power to identify pathological complete response (pCR) in patients with locally advanced rectal cancer after neoadjuvant chemoradiation (nCRT). The model was developed and validated using retrospective cohorts and tested in a multi-institutional prospective trial to demonstrate its generalizability. This model may serve as a noninvasive predictor of pCR after nCRT, potentially providing valuable information in the selection of patients for the organ-preserving strategy.

## Figures and Tables

**Figure 1 bioengineering-10-00634-f001:**
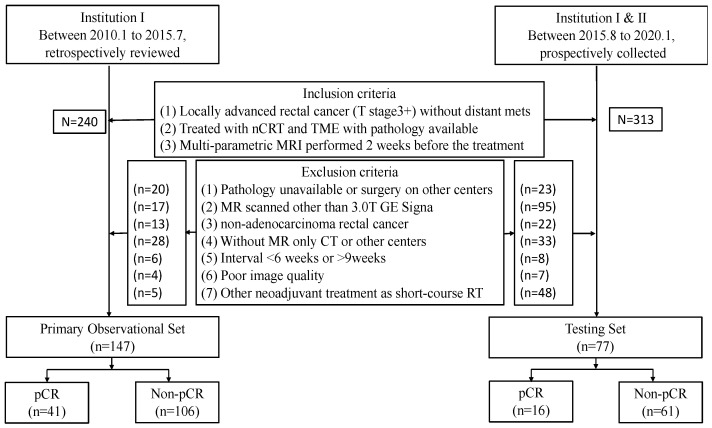
Flowchart summarizing the patient selection process. nCRT, neoadjuvant chemoradiation treatment.

**Figure 2 bioengineering-10-00634-f002:**
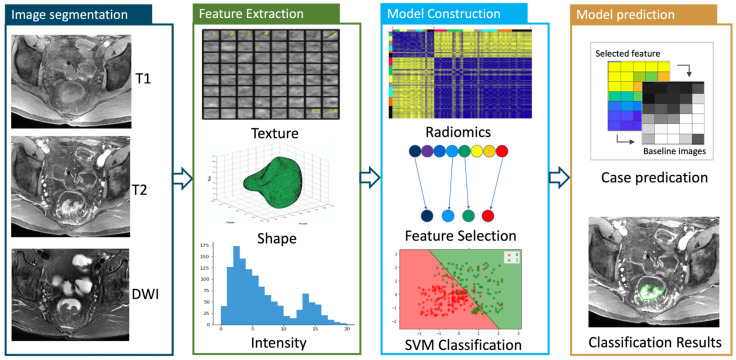
The radiomics workflow chart. The grean part shows the lesion areas on image.

**Figure 3 bioengineering-10-00634-f003:**
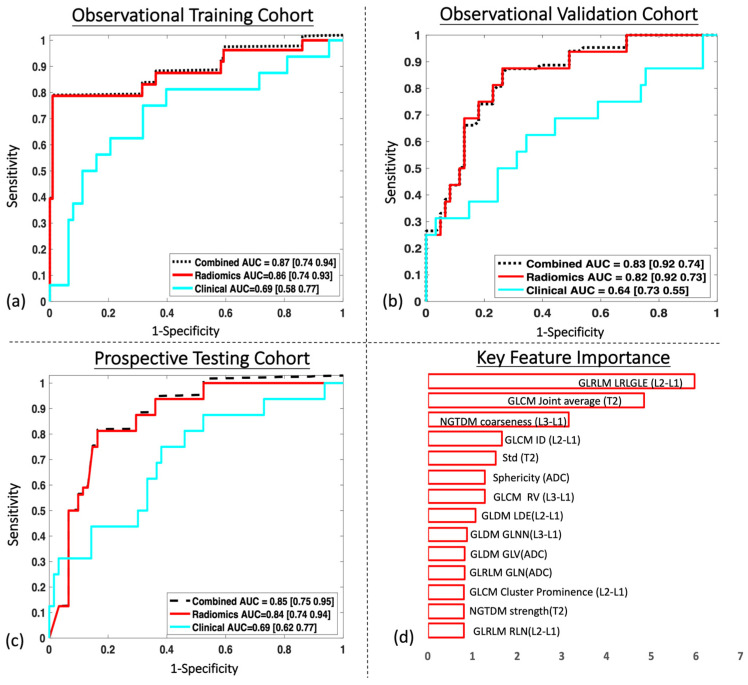
Receiver operating characteristics curve analysis of the radiomics model in: (**a**) the retrospectively collected observational training cohort; (**b**) the retrospectively collected observational validation cohort; (**c**) the prospectively collected testing cohort; and (**d**) key feature importance weightings.

**Figure 4 bioengineering-10-00634-f004:**
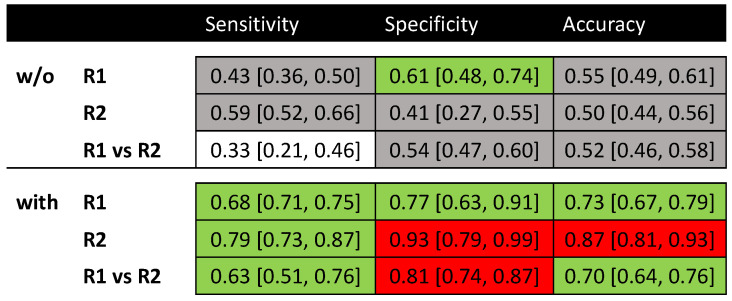
The consistency of intra- and inter-readers’ agreement without (w/o) and with the aid of the radiomics model using the weighted kappa criteria. R1, reader #1; R2, reader #2. (κ = 0.00–0.20, poor and 0.21–0.40, fair (in white); 0.41–0.60, moderate (in gray); 0.61–0.80, good (in green), and 0.81–1.00, excellent (in red) agreement).

**Figure 5 bioengineering-10-00634-f005:**
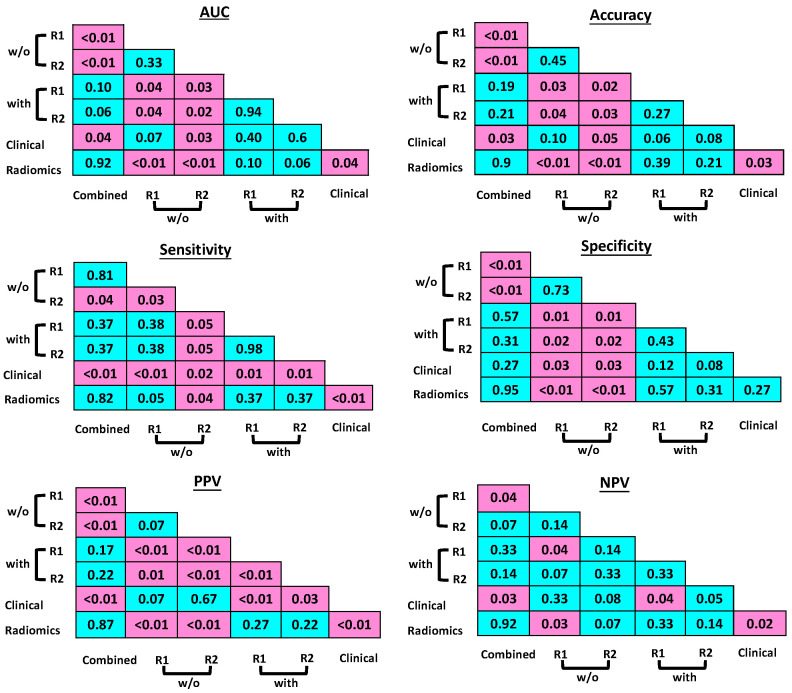
Cross-comparison *p*-values. The Delong test was used to compare AUCs. An exact Cochran–Mantel–Haenszel test (CMH) was used for comparing all other prediction indices with *p* ≤ 0.05 considered to be statistically significant and highlighted in pink. The non-significant results were shown in blue.

**Figure 6 bioengineering-10-00634-f006:**
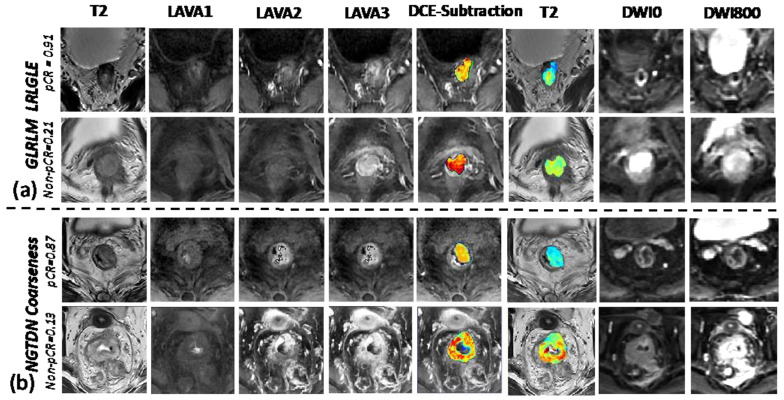
Examples of two features from DCE-MRI subtraction images. Images showing with T2-weighted images, pre-contrast first phase LAVA DCE sequence, later arterial second phase LAVA sequence, venous third phase LAVA sequence, subtraction DCE (2nd-1st LAVA) with color-coded map showing subtraction signal of the tumor, T2w with color-coded map showing T2 signal inside of the tumor, DWI with b = 0 s/mm^2^, and DWI with 800 s/mm^2^: (**a**) GLRLM_LRLGLE, long run low gray level emphasis (LRLGLE) from gray level run length matrix (GLRLM); (**b**) NGTDN coarseness, coarseness measured from neighborhood gray-tone difference matrix.

**Figure 7 bioengineering-10-00634-f007:**
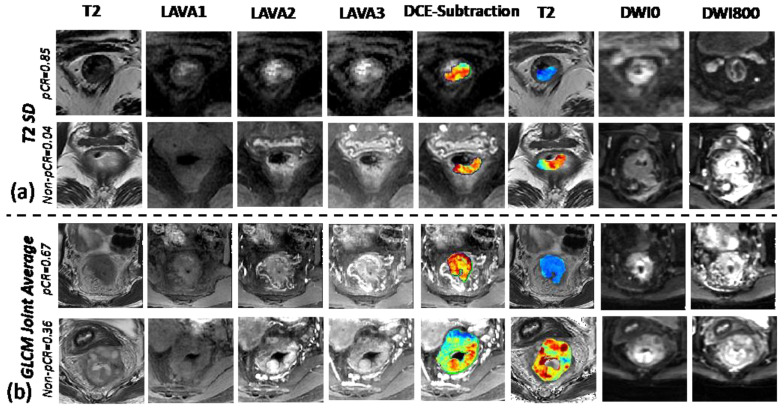
Examples of two radiomic features from T2w MRI. Images showing with T2-weighted images, pre-contrast first phase LAVA DCE sequence, later arterial second phase LAVA sequence, venous third phase LAVA sequence, subtraction DCE (2nd-1st LAVA) with color-coded map showing subtraction signal of the tumor, T2w with color-coded map showing T2 signal inside of the tumor, DWI with b = 0 s/mm^2^, and DWI with 800 s/mm^2^: (**a**) T2 SD, histogram standard deviation from T2w; (**b**) GLCM joint average, joint average from the gray level co-occurrence matrix.

**Table 1 bioengineering-10-00634-t001:** Patients’ characteristics.

Clinical Characteristic	Retrospective Observational Cohort (*n* = 147)	Prospective Testing Cohort (*n* = 77)	*p*-Value
Age (y) *	58.43 ± 10 [27–80]	59.37 ± 9.52 [27–75]	0.49 §
Gender			
Male	104	55	0.98 +
Female	43	22	
Tumor location			
Upper	14	4	0.21 ¶
Middle	93	52	
Lower	40	21	
cT-stage			
3a	80	28	0.05 ¶
3b	44	21	
3c	12	17	
3d	5	5	
4	6	6	
cN-stage			
0	33	6	0.06 ¶
1	74	42	
2	40	29	
Concurrent chemo			
Capecitabine	22	19	0.11 +
Oxaliplatin	125	58	
Adjuvant chemo			
Folfox6	108	67	0.07 ¶
Xelox	27	7	
Capecitabine	8	1	
None	4	2	
Pathology result			
pCR	41	17	0.41 ¶
Non-pCR	106	60	

* Data are means ± standard deviations, with ranges in []. § The Mann–Whitney U test was performed; +, the Fisher’s exact test was performed; ¶, the chi-square test was performed.

## Data Availability

Data that support the findings of the study are available upon request from the corresponding author.

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
