# Peer review of "Radiomics for the Prediction of Pathological Complete Response to Neoadjuvant Chemoradiation in Locally Advanced Rectal Cancer: A Prospective Observational Trial"

_bioengineering, 2023, doi:10.3390/bioengineering10060634_

Round 1

Reviewer 1 Report (Previous Reviewer 4)

This study aims to validate a radiomics model for predicting pathological complete response (pCR) in patients with locally advanced rectal cancer (LARC) who have undergone neoadjuvant chemo-radiation treatment (nCRT). The model was developed and validated on a retrospective cohort of 147 patients and tested on a prospective cohort of 77 patients from two institutions. The radiomics model was constructed using T2-weighted, diffusion-weighted, and dynamic contrast-enhanced MRI and showed better performance than physicians' visual assessment in predicting pCR in the prospective test cohort. With the aid of the radiomics model, junior physicians were able to achieve comparable performance as senior oncologists. This study suggests that a radiomics model based on pre-treatment MRI can be a useful tool for predicting pCR in LARC patients who have undergone nCRT.

Major:

In the background section of the study, one limitation could be the lack of information on the previous studies related to the radiomics model for the prediction of pathological response in LARC patients. This could potentially lead to a knowledge gap and limit the context and understanding of the current study's contribution to the field.

Another limitation in the background section could be the lack of discussion about the potential biases in the patient selection process. For example, the study only included patients scanned with 3.0T GE scanners, which may not be representative of the general population of LARC patients. This could limit the generalizability of the results to other patient populations.

Some limitations of this methodology include:

  1. Limited generalizability: The study was conducted at two specific institutions, and only patients who met specific inclusion and exclusion criteria were included. Therefore, the results may not be generalizable to other populations or settings.

  2. Retrospective design: The observational cohort used in the development and validation of the radiomics model was retrospective, which may introduce bias into the study.

  3. Small sample size: The number of patients included in the study is relatively small, which may limit the statistical power and generalizability of the findings.

  4. Potential measurement errors: There may be variations in image acquisition and segmentation, which can introduce measurement errors into the radiomics features.

  5. Limited clinical variables: The radiomics model was developed based solely on imaging features, without considering other clinical variables that may be relevant, such as patient age, comorbidities, and treatment response.

  6. In the discussion section, one limitation could be the lack of discussion on the potential clinical applications and implications of the radiomics model. While the study shows promising results in predicting pathological response in LARC patients, it is unclear how this information could be utilized in clinical practice to improve patient outcomes.

    Another limitation in the discussion section could be the lack of comparison to other methods or models used for predicting pathological response in LARC patients. This could limit the understanding of how the radiomics model compares to other established methods and the potential advantages and disadvantages of each. Additionally, there could be a lack of discussion on the limitations of the SVM algorithm used in this study and how it may affect the accuracy and generalizability of the model.

  7. To improve the introduction, you could provide more context and background information on CTLA-4 and its role in cancer. For example, you could explain how CTLA-4 inhibits T cell activation and how this can promote tumor growth and metastasis. Additionally, you could discuss the current understanding of the relationship between CTLA-4 expression and colorectal cancer, including previous studies that have examined this association. This will help to better situate the current research within the existing literature and explain the significance of the findings.

    To improve the discussion, you could provide a more detailed interpretation of the results and discuss their implications for future research and clinical practice. For example, you could speculate on the mechanisms by which capecitabine downregulates CTLA-4 expression in SW480 cells and whether this effect is specific to this cell line or more broadly applicable to other colorectal cancer cells. You could also discuss the potential clinical implications of these findings, such as the possibility of combining capecitabine with immune checkpoint inhibitors to enhance the anti-tumor immune response in colorectal cancer patients. Finally, you could acknowledge any limitations of the study and suggest directions for future research to address these limitations and build on the current findings: The findings of the study on the radiomics model for predicting pCR in LARC patients who have undergone nCRT could be discussed in the context of the overexpression of CTLA-4 in CRC tissues. Since the study on the radiomics model was focused on LARC patients who have undergone nCRT, it is likely that some of these patients also had CTLA-4 overexpression in their tumor tissues. The overexpression of CTLA-4 in CRC has been shown to attenuate anti-tumoral immune responses and facilitate tumor growth and metastasis, and it is possible that it may also affect the response to nCRT.

    Therefore, it would be interesting to investigate whether the radiomics model developed in this study could also predict the expression of CTLA-4 in LARC patients who have undergone nCRT. This could potentially lead to a better understanding of the mechanism of action of nCRT in LARC patients and help identify patients who may benefit from additional therapies targeting CTLA-4 expression.

    Furthermore, the use of the radiomics model could potentially help identify patients who are more likely to achieve pCR after nCRT, which is associated with better long-term outcomes. This could aid in treatment decision-making and improve patient outcomes. Additionally, the finding that junior physicians could achieve comparable performance as senior oncologists with the aid of the radiomics model is promising and could have implications for improving patient care and reducing healthcare costs (please refer to 34067631and expand).

  8. Please edit the language.
  9.  

Author Response

Reviewer 2 Report (Previous Reviewer 3)

This revised version is only partly improved compared to the originally submitted paper. I thank the authors for their replies to my comments. However, I suggest to include most of their answers also in the manuscript, which is otherwise unmodified and a reader with my same doubts could not find any answer. Specifically:

1. I didn't find the Supplementary Materials. I believe this is merely a submission error, but I'd like to see them before acceptance.

2. Please add a comment about dataset homogeneity across institutes and the limited impact of ComBat harmonization, even without showing results, to inform the readers about possible batch effects.

3. Regarding the correction for multiple comparisons, please add this detail in the manuscript, and not only on the response to my question.

4. Figure 6 and 7: you have modified colorbars only, but figures are still heterogeneous in crop and background. Please, remove the white box in DCE subtraction and use the same field of view for each image in the same row.

Author Response

Reviewer 3 Report (New Reviewer)

The manuscript was written correctly. Actually, I have one important remark that requires clarification, discussion based on literature findings and authors' thoughts.

Namely, a DWI sequence is used to determine the ADC, i.e. 3 orthogonal DWI measurements from which the ADC diffusion coefficient is calculated. Note that DWI measurements are sensitive to various artifacts, motion, eddy currents, etc. In addition, what seems to be the most important, the values of b and the direction of the diffusion gradient vector show a spatial character, which is the source of systematic errors, precisely of a spatial nature, characteristic for a given MR scanner, the MR sequence used and its parameters. This error can be very large and is strictly dependent on the position of the examined ROI in the laboratory frame of the MR scanner. So, for example, we can observe certain spatial patterns characteristic of different scanners?

This problem is present in the context of issues:

- Systematic Errors in DWI/DTI

- Correction of Errors in DWI/DTI

- Validation of BSD-DTI

- Generalized ST equation for non-uniform gradients

- Phantoms for BSD-DTI

- Anisotropic phantoms for BSD-DTI

- Improving the accuracy of DTI experiments.

This is extensively analyzed in the context of B-matrix spatial distribution in DTI. Here it is important to note that the DTI is based on the individual DWI measurements, so for determining the ADC it is the same situation.

Appropriate discussion showing the potential impact of systematic errors on the results/model is essential.

In addition, improving/providing a more readable potential workflow of model application. What is needed to use the model in everyday clinical practice?

Round 2

Reviewer 1 Report (Previous Reviewer 4)

The authors have clarified several of the questions I raised in my previous review. Unfortunately, some of the major problems have not been addressed by this revision.

The text describes a study that aimed to develop and validate a radiomics model for predicting pathological complete response (pCR) in patients with locally advanced rectal cancer (LARC) who received neoadjuvant chemoradiation treatment (nCRT) and total mesorectal excision (TME). The study involved a retrospective cohort for model development and a multi-center prospective cohort for model validation. The inclusion and exclusion criteria for each cohort are presented, and details regarding the image acquisition and analysis protocols are provided. The radiomics model was constructed using feature extraction, feature selection, and support vector machine (SVM) classification techniques. The model's performance was evaluated using various metrics, including the area under the curve (AUC). The study highlights the limitations of previous radiomics studies and emphasizes the importance of transparent reporting, reproducibility, and multi-center prospective validation to ensure the generalization and clinical applicability of the radiomics model.

The radiomics model was validated in a multi-center prospective cohort that included 77 patients from two institutions (am I wrong?). The patients in this cohort were enrolled in a prospective trial and received the same treatment protocol as the patients in the observational cohort. The radiomics model developed in the observational cohort was applied to the pre-treatment MRI images of the patients in the testing cohort, and the prediction performance was evaluated using the AUC and other metrics. The model was considered to have good generalization ability if it achieved similar performance in the testing cohort as in the observational cohort. The multi-center prospective validation is highly recommended to ensure the reliability and generalizability of the radiomics model in clinical practice.

Multi-center prospective validation is an essential step in the development and evaluation of radiomics models for clinical applications. It involves testing the model's performance on independent datasets collected from different institutions using similar but not identical protocols. This approach can help assess the model's generalization ability and its potential to perform well in different clinical settings.

In the study described in the text, the multi-center prospective cohort was used to evaluate the radiomics model's performance on an independent dataset collected from two different institutions. The model was trained using the observational cohort from Institution I and validated using the testing cohort from Institution I and Institution II. The patients in the testing cohort were subjected to the same inclusion and exclusion criteria as those in the observational cohort, and the same treatment protocol was followed.

The performance of the radiomics model was evaluated using various metrics, including the AUC, sensitivity, specificity, and accuracy. The results showed that the model achieved good performance in both the observational and testing cohorts, indicating its generalization ability. The authors emphasized the importance of multi-center prospective validation to ensure the reliability and generalizability of the radiomics model in clinical practice.

The text does not provide a detailed discussion of the limitations of the methodology used in the study. However, some potential limitations can be inferred from the information provided.

One limitation of the study is that the radiomics model was developed using a retrospective cohort design, which may introduce bias and limit its generalizability to other patient populations. Although the model was validated in a multi-center prospective cohort, it would be ideal to develop the model using a large, multi-center prospective study design to ensure its generalizability and clinical applicability.

Another limitation is that the radiomics model was developed using a specific MRI protocol (3.0T GE scanners with T2w, DCE-MRI, and DWI sequences), which may not be available in all clinical settings. The model's performance may vary when applied to images acquired using different protocols or from different imaging modalities.

The study also used a relatively small sample size, which may limit the statistical power of the analysis and increase the risk of overfitting. A larger sample size would provide more robust results and improve the model's generalizability.

Finally, the study did not provide information on the reproducibility of the radiomics features used in the model or the inter-observer variability of the ROI segmentation. These factors can affect the model's performance and limit its clinical applicability.

The authors slightly expanded the section regarding CTLA4 discussion, nevertheless, no references are provided (in the previous review round this reviewer suggested referring to PMID: 340676631 or alternative manuscripts.) 

Author Response

Reviewer 2 Report (Previous Reviewer 3)

The authors have answered to all the points that I raised in the previous revisions. I'm now satisfied with the current manuscript.

Author Response

We are truly grateful for the time and effort you have invested in assisting us to enhance the quality of our paper.

Reviewer 3 Report (New Reviewer)

In general, the manuscript, as I wrote, was well written, now undoubtedly even better, nevertheless, there is still a significant lack in the matter raised.

Namely, the statement contained by the authors is true as well as the discussion of possible consequences for the considerations contained in the manuscript. However, the lack of any reference to specifics, here it can be done by giving specific items in the literature, makes this sentence incomprehensible and irrelevant.:

"Additionally, factors such as gradient nonlinearity, magnetic field inhomogeneity, motion artifacts, and eddy current-induced distortions can result in discrepancies and artifacts within DWI images." .

Therefore, I propose to add specific literature items, in the context of heterogeneity of magnetic field gradients, we have 2 main paths. Coil tensor L proposed by Bammer in 2003, where we assume that inhomogeneities come only from gradient coils. And the BSD-DTI trend, B matrix Spatial Distribution in DWI/DTI, where the diffusion tensor standards are used to determine the spatial distribution of the b matrix, which is a function of the spatial distribution of field gradients. And here it is possible to determine the actual distribution of field gradients, including those coming from gradient coils, but also background, eddy currents or radiation damping. I suggest at least one reference in each of these 2 trends and provide the consequences: systematic errors in DWI/DTI, "correction of errors in DWI/DTI" along with references.

That's the bare minimum for it to make sense and to be readable by redears.

Author Response

This manuscript is a resubmission of an earlier submission. The following is a list of the peer review reports and author responses from that submission.

Round 1

Reviewer 1 Report

1.         The sample size of the study is relatively small, which may not be representative of the larger population.

2.         The study only includes two institutions, which may limit the generalizability of the results.

3.         The study only uses MRI as the imaging modality, which may not be the most accurate or feasible option for all patients.

4.         The study does not address the cost-effectiveness of implementing a radiomics model in clinical practice.

5.         The study does not evaluate the long-term outcomes or survival rates of patients who were predicted to have a pathological complete response using the radiomics model.

6.         The study does not compare the radiomics model with other established methods for predicting pathological complete response.

7.         The use of a retrospective cohort for model development and a prospective cohort for validation may introduce bias and limit the generalizability of the results.

8.         The study is limited by the lack of a true gold standard for pCR assessment, which raises questions about the accuracy and reliability of the radiomics model.

9.         The results of the study should be interpreted with caution as the radiomics model is only one aspect of the evaluation and it is not clear if the model can be used for clinical decision making.

10.     The study design is limited by the fact that it only includes patients who were scanned with 3.0T GE scanners (Signa) from two centers, which may not be generalizable to other patient populations or MRI systems.

11.     The study focuses on a single imaging modality (MRI) and specific MRI sequences, which may not fully capture all aspects of the disease.

12.     The study relies on the interpretation of pathology specimens by multiple pathologists, which may introduce bias and variability in the results.

13.     The study does not provide a detailed description of the radiomics model, making it difficult to assess its accuracy and reproducibility.

14.     The study does not compare the performance of the radiomics model with other commonly used methods for pCR prediction, such as clinical and pathological factors.

15.     The study only includes patients scanned with a specific type of scanner, which limits the generalizability of the findings.

16.     The radiomics process is not well described and it is not clear how the specific features and parameters were chosen.

17.     The sample size of the prospective cohort is relatively small, which may limit the statistical power of the study.

18.     The use of synthetic minority oversampling technique may introduce bias and affect the generalizability of the model.

19.     The study only compares the radiomics model to a clinical model, and does not compare

20.     The sample size of the study is relatively small, with only 77 patients in the prospective cohort.

21.     There is no mention of how the radiomics model was validated in a clinical setting or how it would be implemented in clinical practice.

22.     The study does not compare the radiomics model to current standard of care methods for predicting pCR in LARC patients.

23.     There is no mention of how the model would perform in a diverse patient population and whether it would generalize to different ethnic or racial groups.

24.     The study does not discuss the cost-effectiveness of implementing the radiomics model in clinical practice.

Author Response

We would like to thank editor, associate editor and all reviewers for their time and effort in helping us to improve the quality of the paper. All comments have been addressed in BOLD accordingly.

Reviewer-1

  1. The sample size of the study is relatively small, which may not be representative of the larger population.

Thanks. Agreed. We have listed as a limitation of the study. “.... Firstly, sample size was still limited, which was a general concern for most of the radiomics based studies. The study was conducted within two institutions, using the same strength MRI scanners and identical image protocols, which may limit the generalizability of the results. This is an observational study and recruitment of larger populations under random trials is needed for further analysis. Although beyond the scope of this observational study, cost-effectiveness analyses are also needed to adapt the model for direct clinical use.”

  1. The study only includes two institutions, which may limit the generalizability of the results.

Thanks. We have listed as a limitation of the study. Please see response #1.

  1. The study only uses MRI as the imaging modality, which may not be the most accurate or feasible option for all patients.

Thanks. We have listed as a limitation of the study. Please see response #1.

  1. The study does not address the cost-effectiveness of implementing a radiomics model in clinical practice.

Thanks. We have added “... Although beyond the scope of this observational study, cost-effectiveness analyses should be conducted to adapt the model for direct clinical use...”

  1. The study does not evaluate the long-term outcomes or survival rates of patients who were predicted to have a pathological complete response using the radiomics model.

The aim for this study is to focus on early prediction of pathological complete response (pCR). It is reported that patients with pCR after nCRT have general favorable oncological prognosis, with a 5-year disease-free survival rate reaching 83%-95%. Additionally, the observation of pCR has led to a paradigm shift in rectal cancer management. To avoid surgery-associated morbidity and impairment in quality of life, patients with complete response following nCRT are increasingly being offered watch-and-wait regimens or organ-sparing strategies, such as local excision. To further increase the number of eligible patients for such organ preservation strategies, physicians are searching for (new) neoadjuvant treatments with higher organ-sparing potential than the current standard of care, as well as searching for potential biomarkers to predict pathological complete response before the surgery. We have added this background information into Discussion.

  1. The study does not compare the radiomics model with other established methods for predicting pathological complete response.

Thanks. Currently, there is no established model or method to predict pathological complete response and relevant research is limited. Only selective clinical parameters were reported to be associated with pathological complete response, which includes TNM-staging, tumor size, tumor location etc. Moreover, most of the parameters reported in the literature were conducted in research settings and not verified by large clinical trials. Thus, we compared the results with clinical models that contain clinical parameters captured during clinical settings.

  1. The use of a retrospective cohort for model development and a prospective cohort for validation may introduce bias and limit the generalizability of the results.

Thanks. Table 1 provided patient characteristics between the retrospective cohort and prospective cohort, showing no statistical differences between the two groups.  We also performed power analysis based on the retrospective cohort findings. “...The sample size of prospective cohort was calculated based on the accuracy of radiomics model in predicting pCR from the retrospective cohort with an assumption of 15-20% pCR incidence rate and a goal of alpha= 0.05, 1-beta =0.95 using SPSS 26...”, “...Power analysis based on retrospective observational cohort results revealed that a minimum of 65-72 patients were needed to achieve reasonable power. In the prospective cohort of the study, we have collected 77 patients with the same inclusion and exclusion protocol to meet the power requirement. On the other hand, the fact that the prediction accuracy showed comparable results with the retrospective cohort in this observational cohort. Again, studies with larger population are needed to verify the generalization of this model.”

  1. The study is limited by the lack of a true gold standard for pCR assessment, which raises questions about the accuracy and reliability of the radiomics model.

The gold standard for pCR assessment is from the pathological reading which can only be confirmed after surgery. The aim in early identification of pCR is the potential benefit of avoiding extensive surgery, which has a significant risk of perioperative morbidities, including bowel, sexual and urinary dysfunction. Unfortunately, there is no established model or method to predict pathological complete response. Only selective clinical parameters were reported to be associated with pathological complete response, which includes TNM-staging, tumor size, tumor location etc. Moreover, most of the parameters reported in the literature were conducted in research settings and not verified by large clinical trials. Thus, we compared the results with clinical models that contain clinical parameters captured during clinical settings.

  1. The results of the study should be interpreted with caution as the radiomics model is only one aspect of the evaluation and it is not clear if the model can be used for clinical decision making.

Thanks. We have listed as a limitation of the study. Please see response #1.

  1. The study design is limited by the fact that it only includes patients who were scanned with 3.0T GE scanners (Signa) from two centers, which may not be generalizable to other patient populations or MRI systems.

Thanks. Please see response #1.

  1. The study focuses on a single imaging modality (MRI) and specific MRI sequences, which may not fully capture all aspects of the disease.

Thanks. Please see response #1.

  1. The study relies on the interpretation of pathology specimens by multiple pathologists, which may introduce bias and variability in the results.

Thanks. The clinical endpoint in this study is based on pathological report reading instead of TRG staging which reduces the bias in clinical readings. Additionally, the pathological reading was verified by an experienced pathologist with minimal 10-year experience.

  1. The study does not provide a detailed description of the radiomics model, making it difficult to assess its accuracy and reproducibility.

We have added more details regarding the construction of the radiomics model, please refer to Appendix #4.

  1. The study does not compare the performance of the radiomics model with other commonly used methods for pCR prediction, such as clinical and pathological factors.

Please see response to #8.

  1. The study only includes patients scanned with a specific type of scanner, which limits the generalizability of the findings.

Please see response #1.

  1. The radiomics process is not well described and it is not clear how the specific features and parameters were chosen.

We have added more details regarding the construction of the radiomics model, please refer to Appendix #4.

  1. The sample size of the prospective cohort is relatively small, which may limit the statistical power of the study.

Please see response #7.

  1. The use of synthetic minority oversampling technique may introduce bias and affect the generalizability of the model.

Thanks. The Synthetic Minority Over-sampling Technique (SMOTE) is a conventional method for oversampling the minority class in a dataset with imbalanced class distribution. It has been adopted for processing of medical data. It works by generating synthetic samples of the minority class by interpolating between existing samples. To avoid over-representation, we compared the performance of SMOTE and other methods, including random oversampling or cost-sensitive learning etc. However, those methods introduced more bias with worse performance. To find the best hyper-parameter and strategy to control overfitting in this study, we kept assessing the performance of the model on validation data to determine its generalizability through cross-validation.

  1. The study only compares the radiomics model to a clinical model, and does not compare

Please see response to #8.

  1. The sample size of the study is relatively small, with only 77 patients in the prospective cohort.

Please see response #7.

  1. There is no mention of how the radiomics model was validated in a clinical setting or how it would be implemented in clinical practice.

Please see response #1.

  1. The study does not compare the radiomics model to current standard of care methods for predicting pCR in LARC patients.

Please see response to #6 and #8.

  1. There is no mention of how the model would perform in a diverse patient population and whether it would generalize to different ethnic or racial groups.

Thanks. Please see response #1.

  1. The study does not discuss the cost-effectiveness of implementing the radiomics model in clinical practice.

Please see response to #4.

Reviewer 2 Report

Authors validated a radiomics model for pathological complete response assessment on a prospective trial to support informational insight of radiomics validation. They included 147 consecutive patients for development of a radiomics model, and a prospective cohort of 77 patients from two institutions to test its generalization.

The radiomics model outperformed both physicians’ visual assessment in the prospective test cohort
with an area under the curve (AUC) of 0.84 . This is sensational result and , even while this article is relatively difficult from the methodological perspective , it contains several very good images which can used for medical lectures in the future.

I recommend to accept in present form.

Author Response

Reviewer-2

Authors validated a radiomics model for pathological complete response assessment on a prospective trial to support informational insight of radiomics validation. They included 147 consecutive patients for development of a radiomics model, and a prospective cohort of 77 patients from two institutions to test its generalization.

The radiomics model outperformed both physicians’ visual assessment in the prospective test cohort
 with an area under the curve (AUC) of 0.84. This is sensational result and, even while this article is relatively difficult from the methodological perspective, it contains several very good images which can be used for medical lectures in the future.

I recommend to accept in the present form.

Thank you very much for the comments.

Reviewer 3 Report

This work proposed a radiomic analysis from multi-parametric MRI images acquired in a population of rectal cancer patients treated with neoadjuvant chemoradiotherapy, to validate the radiomic model in a prospective cohort and to evaluate the additional value of this model on expert assessment.

The paper is interesting, well written and methodologically robust. From my point of view, the most interesting result is about the use of the radiomic model, that help junior reader in increasing their performance.

To improve the paper, I suggest some modifications or additional analyses. Specifically:

11. Radiomic processing: why did you chose a fixed bin number of 25 to compute texture matrices? It is reported in several studies that radiomic features reach higher reproducibility when computed using a fixed bin number between 32 and 64. Please, justify this choice or correct the analysis.

22. Since the prospective cohort comes from two different institutions, have you evaluated if significant differences are present between the two populations? It is true that you have chosen patients acquired using the same scanner, but batch effect could be anyway present. I suggest to verify this point and perform harmonization between radiomic features (maybe using Combat method) if batch effect is present.

33. Results: I suggest reporting the values (mean and standard deviation, or median and IQR) of the most relevant features in the two groups (pCR and non-pCR), and the final radiomic model used to help the readers.

44. Results, line 195: I didn’t find Table 2 in the manuscript. Please correct or provide the table.

55. Results, Figure 5. Have you corrected these p-values for multiple comparison? Several shown p-values are very closed to the statistical significance threshold.

66. Results, Figure 6 and 7. I suggest you to modify these figures to improve the readability. Why is there a white box surrounding DCE subtraction? Some images are cropped (e.g. T2 of the last row in Figure 6). Please, be consisted in the MRI representation. Finally, the colorbar on the right can confuse the reader. I suppose that this colorbar is only helpful to report the radiomic index in the two patients (pCR and non-pCR), but it could be easily confused for DCE or T2 colormap. I suggest to report the value of the radiomic feature only as numeric value, without colorbar.

Author Response

We appreciate your time and effort. It seems like to the comments may be related to another manuscript with immunotherapy for colorectal cancer cells. Please see the attachment.
